# Dialogue with the public: A catalyst for professional identity formation in medical students

**Masashi Ikuno** ⊙\*, **Akiko Tokinobu** ⊙, **Tomoko Miyoshi, Hitomi Kataoka**

Center for medical education and internationalization, Kyoto University, Kyoto, Japan

\* mikuno@kuhp.kyoto-u.ac.jp

## Abstract

### Objectives

This study aimed to explore how Japanese medical students are influenced by their dialogue with the general public as part of the process of professional identity formation. These findings are expected to provide insights into supporting the development of professional identity from diverse perspectives, particularly for medical students in the pre-stage of legitimate peripheral participation and those currently engaged in it.

### Methods

Semi-structured interviews were conducted with three medical students of different backgrounds. These students participated in a dialogue-based event with the general public, in which they ran a booth focused on medical education. A qualitative descriptive approach was adopted and an interpretivist study was conducted through a thematic analysis. Data were analyzed using the Steps for Coding and Theorization (SCAT) qualitative data analysis method.

### Results

The students reflected on their dialogue with the general public about medical education and discussed infrequent opportunities to recognize or disclose their professional identity in daily life. A preclinical student (third-year) noted, "I felt like I shared a similar sense of amazement with the visitors" and "It's obvious, but I really came to realize that I'm seen from the perspective of a medical student or a healthcare professional." In contrast, a clinical student (fifth-year) reflected, "Sharing my feelings about whether I've been helping others during my clinical training felt closely tied to me—there wasn't much of a gap." SCAT analysis revealed that preclinical students emphasized 'reaffirming their professional identity through empathetic understanding,' while clinical students focused on 'disclosing their professional identity, leading to its integration with their personal identity.'.

**Data availability statement:** All relevant data are within the article and its supporting information files.

**Funding:** Grant from the Ministry of Education, Culture, Sports, Science, and Technology [H.K, Grant-in-Aid for Scientific Research (C), JP24K13315]. The funder had no role in the study design, data collection and analysis,

decision to publish, or preparation of the manuscript.

**Competing interests:** The authors declare no conflicts of interest in association with the present study.

## Conclusion

Dialogue with the general public unexpectedly enhanced medical students' awareness of their professional identity. In the process of professional identity formation, dialogue with the general public can support the traditionally emphasized socialization process through participation in professional groups, particularly by facilitating the reaffirming of professional identity and its integration with personal identity.

## Introduction

Professional identity formation (PIF) is essential for medical students as they grow and prepare to become future physicians. Well-developed PIF ensures that medical students not only acquire the necessary knowledge and skills, but also internalize the ethical values and professional behaviors essential for patient-centered care and lifelong learning as physicians [1,2]. Cruess et al. argued that PIF in medical education occurs through participation in communities of practice (CoPs), progressing from legitimate peripheral participation to full participation [3]. This process of being accepted into the medical community is referred to as "socialization," during which personal identity integrates with professional identity through the development of a sense of professional solidarity. Factors influencing socialization include role models, mentors, and clinical/non-clinical experiences as core elements, along with the influence of the learning environment, healthcare systems, and other aspects [3].

Cruess et al. also highlighted the influence of the general public, stating,' Patients, colleagues, families, other healthcare professionals, and the general public tend to see medical students and residents as members of the broader medical profession. This perception may profoundly influence self-recognition as they progress toward becoming professionals." However, their report did not elaborate on the unique effects of dialogue with the general public.

Cruess et al.'s study provides valuable insights into professional identity formation, but the influence of dialogue with the general public was not a primary focus of this study, which could be partly explained by the timing of the initiation of medical education, which varies by country. For example, in Germany, students begin a six-year medical program immediately after graduating from high school, whereas in the United States, students enter medical school after obtaining a bachelor's degree [4]. In the U.S., medical education occurs after students interact with the general public. In contrast, in Germany and Japan, students often enter medical school directly after high school, meaning that the entire process of identity formation over six years is encompassed within the medical education curriculum. Given this context, dialogue with the general public may be even more critical in countries such as Germany and Japan, where medical education begins directly after high school.

Thus, PIF is traditionally understood as a process of socialization within professional communities, particularly through interactions with mentors, peers, and clinical experiences. While these internal factors have been extensively studied, PIF is also influenced by broader social interactions, including engagement with the general

public. Recent studies suggest that broader social interactions, including engagement with the general public, also contribute significantly to PIF [5,6]. These reports showed that public engagement can provide opportunities to articulate professional roles, receive external validation, and reflect on their evolving identities. However, as mentioned earlier, most existing literature on PIF has focused on socialization within medical communities, leaving a gap in understanding how external interactions shape identity development. This study aims to address this gap by investigating how dialogue with the general public influences the PIF process among Japanese medical students.

To explore this, we conducted a qualitative study using semi-structured interviews with medical students who participated in an interactive public event at Kyoto University. During the event, titled "The Past and Future of Medical Education," students introduced the public to the types of classes and training involved in medical education. By analyzing students' reflections on this outreach activity, we examined how public engagement complements traditional forms of professional socialization. A thematic analysis was conducted using Steps for Coding and Theorization (SCAT).

## Methods

### Study design, participants, and setting

This was a qualitative investigation focusing on the impact of dialogue with the general public on medical students. This research utilized an outreach event regularly held by Kyoto University, "Academic Day 2024." The study was targeted all three students who participated in running a presentation booth on medical education during Academic Day 2024. These students included one third-year female student with no clinical training experience, and two fifth-year students (one male and one female) with clinical training experience. All student participants were selected using convenience sampling. The recruitment period for this study was from September 17 to October 31, 2024. The total study period lasted three weeks, covering the time from the public interaction event to the completion of all interviews and data processing. The public interaction event itself took place on a single day, lasting five hours. After the event, individual interviews were conducted once per participant. Participants were provided with an explanation of the research content in person, along with a written explanation document. Consent was obtained in accordance with the IRB-approved format and procedure through a web form. Participants were required to review the consent document, check an agreement checkbox, and enter their name to confirm their consent. This study was approved by the Kyoto University School of Medicine Ethics Committee (approval number: R4467).

### Details of the dialog with general public event

Academic Day at Kyoto University is an initiative aimed at providing a platform for "dialogue," enabling everyone to discover the joy and appeal of academic pursuits. Held annually, the event gathers more than 100 researchers in one place, offering them the opportunity to engage directly with the public and explain the university's research and educational activities in an accessible manner. Additionally, it facilitates the incorporation of public feedback into universities' research activities. From 2011 to 2023, the event was jointly planned and managed by the "Dialogue with Society on Science and Technology" working group, comprising members from the Kyoto University Research Administration Center (KURA), the Research Promotion Division, and Kyoto University faculty and staff. Since 2024, KURA has taken the lead in organizing the event, focusing on creating a space for dialogue that transcends boundaries within and outside the university. In 2024, the event was held twice: once in September and once in November. This study focused on the November event held at Kyoto University's premises. During this event, the principal investigator, together with the students, ran a booth showcasing posters on the evolution of medical education and simulators or VR equipment used for participatory clinical training. The original content of the medical education-related posters used by the students for dialogue with the general public is presented in Fig 1, and their English translations are provided in Figs 2 and 3. The simulators and tools displayed, along with the interactions facilitated through them, are presented in Table 1. The event lasted for five hours, from 11:30 A.M. to 16:30 P.M. Throughout this period, the booth received a constant flow of visitors, and the students engaged in continuous dialogue with them, except during the designated break times.

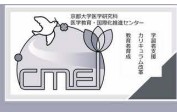

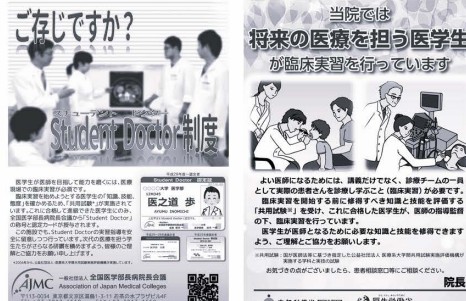

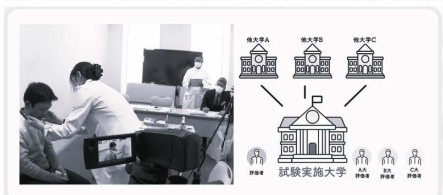

**Fig 1. Original poster of the event (Japanese).**

## The Present and Future of Medical Education
### Center For Medical Education and Internationalization

### From Observational to Participatory Clinical Training

Traditional clinical training in Japan, known as 'polikuri,' is observational. The issue of students obtaining medical licenses and immediately starting work as residents without having performed any medical procedures was raised, and discussions advocating for participatory clinical training began in the 1970s. It has taken many years to establish the legal framework necessary for this approach.

**April 2023: Implementation of the Revised Medical Practitioners' Act** Medical procedures performed by medical students during clinical training are now legally permitted.

**Emergence of the Concept of 'Student Doctor'**

**Revised Medical Practitioners' Act**
The revised act was enacted in May 2021, with the addition of Article 17-2. It came into effect on April 1, 2023.

**Article 17**
No person other than a licensed physician shall engage in medical practice.

**Article 17-2**
Notwithstanding the provisions of the preceding article, students majoring in medicine at universities who have passed a common examination prescribed by the Ministry of Health, Labour and Welfare ordinance, which is conducted by universities to evaluate whether the students have acquired the knowledge and skills that must be learned before starting clinical training, may engage in medical practice during clinical training conducted by their universities under the guidance and supervision of physicians, for the purpose of acquiring the knowledge and skills required of physicians.

Medical Procedures That Medical Students Are Permitted to Perform (Excerpt)(Based on the Ministry of Health, Labour and Welfare's report: "Research on Medical Procedures Feasible in Clinical Training in Medical Schools," published July 31, 2018)

**Required Procedures:**
Documentation of medical records; medical interviews; vital sign checks physical examination techniques; skin disinfection; venipuncture injections (subcutaneous, intradermal, intramuscular, intravenous); skin suturing; hemostasis procedures; ultrasound examination (cardiovascular); ultrasound examination (abdominal); basic life support (BLS); airway management; development of treatment plans

**Recommended Procedures:**
Explanation of medical conditions to patients and families
Incision and drainage of abscesses
Wound care, burn treatment, defibrillation
Endotracheal intubation, orthopedic conservative treatments such as fixation

**Fig 2. Upper half of the poster (Translated to English).**

## Interviews and data collection

Interviews were conducted with students who participated in running the medical education booth during the Academic Day in Kyoto University 2024. Individual face-to-face interviews were conducted with each student. Before the interviews, participants were informed that their participation was entirely voluntary. They were explicitly informed of their right to refuse participation, withdraw at any time without providing a reason, or pause or terminate the interview midway. The interviews were conducted by the principal investigator, who was not involved in teaching the courses taken by the students or in their academic evaluations, ensuring that their progression had no influence on their decision. Before the interviews, it was explained that the study aimed to investigate the potential impact of their participation in the event as part of medical education research. Participants were encouraged to report any negative experiences, as doing so would enhance the quality of the study. These considerations aimed to ensure that respondents could answer questions without fear or anxiety, and to minimize biases arising from power dynamics during the interview process.

**Fig 3. Lower half of the poster (Translated to English).**

An interview guide was prepared and used to maintain a consistent structure throughout the interviews (Table 2). The guide was developed through discussions among all authors of the study. Each interview lasted 15–30 minutes and was conducted in a private room where there was no risk of being overheard by others. The seating arrangement was such that the interviewee was positioned closer to the door, allowing them to leave easily, if they desired. Audio recordings of the interviews were collected by the principal investigator, saved in the MP3 format, and transcribed by the first author.

## Data analysis

Audio recordings of the interviews were promptly transcribed by the first author. The transcribed data were then analyzed using the Steps for Coding and Theorization (SCAT) qualitative analysis method. SCAT is a four-step coding process for qualitative data analyses developed by T. Otani of Nagoya University. This method is used to identify themes and structures, weave them into a narrative, and present a theory [7,8]. SCAT has been utilized in fields such as medical education and palliative care [9–14]. The analysis consists of two main steps. In the first step, keywords were extracted from the context, replaced with technical terms, and grouped into themes. In the second step, the generated themes were used to reconstruct a narrative, leading to theorization. This clear analytical process ensures that the findings are open to critique and refutation, thereby guaranteeing the validity of the theory.

**Table 1. Simulators and tools in the event.**

| Simulator Name | Overview | Usage on the Day of the Event |
|---|---|---|
| Appakun® | A simple simulator for training in chest compressions, heart massage, and AED pad placement. It emits a sound when chest compressions are performed correctly. | Primarily, elementary school visitors showed interest, and medical students taught children how to perform heart massage while explaining the role of basic life support (BLS) in the medical education curriculum. |
| Shinjō ®(Blood Sampling Simulator) | A simulator for training in blood sampling techniques. | For safety reasons, needles were not used on the event day. Visitors were encouraged to touch the simulator and locate the veins hidden beneath the surface. Medical students explained techniques for puncturing invisible veins and shared that, in the past, when such simulators were unavailable, students were required to practice on each other. |
| Choking Charlie® (Heimlich Maneuver Training Device) | A simulator for practicing the Heimlich maneuver in cases of foreign body aspiration. | Mainly high school boys showed interest and repeatedly attempted the maneuver until the cotton ball representing the foreign body was expelled. Medical students, along with senior students guiding junior students, prepared for the event, and the atmosphere was lively with applause when visitors succeeded. |
| Meta Quest 2 (VR Goggles)® | Loaded with software developed in collaboration with companies by the Kyoto University Center for Medical Education and Internationalization, visitors experienced simulations. | Specifically, the content allowed simulation of chest examinations, abdominal examinations, and neurological examinations. The feature that gained particular attention was the simulation of tendon reflexes, where real tendon hammers were linked to VR, allowing visualization of muscles and tendons in the virtual space. Young visitors enjoyed the experience, while older attendees asked questions about the modern methods of medical education. Medical students explained the similarities and differences between VR-based practice and clinical training. |
| Edema Simulator | A simulator worn on the shin to examine various levels of pitting edema. | A simulator worn on the shin to simulate various levels of pitting edema. Some general visitors expressed surprise at the severity of the condition, which exceeded their expectations. Medical students shared their clinical training experiences, including encounters with patients suffering from severe edema. |
| Examination Tools (Ophthalmoscope, Reflex Hammer, Tuning Fork, Toothpick) | Displayed tools commonly used in routine examinations, particularly in neurological assessments. | Visitors were free to handle the tools, and medical students explained their uses. Elementary school visitors were particularly surprised that everyday items like toothpicks are used in neurological exams. This often served as a good conversation starter between the students and the visitors. |

**Table 2. Interview guide.**

| |
|---|
| **1. Feelings When Deciding to Participate in Academic Day 2024** |
| · What motivated you to participate in the event? |
| · What factor/s made you decide to participate? |
| **2. Impressions of Participation** |
| · What were your feelings after participating upfront on Academic day? |
| · What were the positive versus challenging aspects of participation.? |
| **3. Communication** |
| · Through this experience, did you gain any new insights or awareness about communication? |
| **4. Personal Changes** |
| · Did this experience bring any changes or enhancements in your motivation for studying and training in medical school? |
| **5. Teamwork** |
| · What are your thoughts on working together with other students during this event? |
| **6. Past Experiences in Dialogue with the General Public** |
| - Did you have any previous opportunity/ies to engage in a dialogue with the general public like this? |
| · Can you share any experience/s of discussing the medical curriculum or medical practices with friends, family, or others before? |

## Results

This study involved three participants. Each participant was interviewed once following their participation on Academic Day 2024. The average interview duration was 21 min. A thematic analysis using SCAT was conducted for each individual interview, and theoretical descriptions were categorized and examined based on their content. The following effects on students were identified:

The students reflected on their dialogue with the general public about medical education and demonstrated the following positive changes:

**All students:** "Rarely experiencing awareness or disclosure of their professional identity in everyday life."

**Third-year student (junior student in the earlier phase of their medical education):**

"Reaffirming their professional identity through empathetic understanding of visitors."

**Fifth-year student (senior student in the advanced stage of clinical training):**

"Disclosing their professional identity during dialogues with the general public, leading to the integration of professional identity and personal identity."

These detected effects were based on the statements provided by the participants, as detailed below.

### Limited awareness or disclosure of professional identity in daily life

Students mentioned that they rarely had opportunities to touch upon their identities as medical students in interactions within their personal lives. This was attributed to the following factors:

- Among medical students, their shared identity is taken for granted and thus remains invisible.

- In interactions with students from other faculties or family members, discussion about medical education tends to be avoided, as it involves considerable effort to explain but does not lead to engaging conversations.

- There is a sense of anxiety that mentioning their identity as medical students might be perceived by others as boastful.

   5MS "Even if I talk about medical education, (...) it's just not that interesting."
   5FS "People kind of pull back, and then it feels like I can't bring up that I'm in medical school anymore."
   3FS "I don't really want to make a big deal about how hard it is, so I just don't bring it up."

### Reaffirmation of professional identity through empathetic understanding of visitors

The third-year student, who had not yet experienced clinical training, spoke about the empathetic understanding they had developed toward the general public as a preclinical student and described how this dialogue reaffirmed their own professional identity.

Before preparing for the dialogue-based event in this study, the third-year students had no prior knowledge of the evolution of medical education or simulators used for clinical training. Consequently, she was able to engage in conversations from a perspective that was similar to that of the general public. Interestingly, this allowed her to discover that she was still perceived as a medical student by the public despite this shared perspective.

   3FS "I felt like I shared a similar sense of amazement with the visitors."

   3FS "It's obvious, but I really came to realize that I'm seen from the perspective of a medical student or a healthcare professional."

### Disclosure of professional identity through dialogue with the general public and integration of professional and personal identity

Fifth-year students who had clinical training experience reflected on how dialogue with the general public prompted them to revisit their peripheral participation in the community of practice. They also shared that through natural conversations

based on their practical experiences, they felt a seamless integration of their personal identity with their identity as medical students.

The students mentioned that in their daily lives, they rarely had opportunities to talk about their medical education and efforts to become physicians, even with their parents or close acquaintances. However, during the dialogue with the general public, they found themselves naturally discussing their efforts. This experience helped them embrace their identities as medical students striving to become physicians.

5MS "Even though I talked a lot, it did not feel mentally exhausting because I was speaking based on my experiences rather than overthinking."

5FS "I realized I could now talk about the efforts I'm making to become a doctor, which felt natural."

5FS "Sharing my feelings about whether I've been helping others during my clinical training felt closely tied to me—there wasn't much of a gap."

5FS "Saying, 'I'm working hard to become a doctor' just feels right now—it resonates with me."

## Discussion

PIF among medical students is often conceptualized as a process of socialization through participation in professional communities, as exemplified by the framework of Cruess et al. [3]. This process is influenced by various factors, but the impact of dialogue with the general public—individuals who do not belong to the professional medical community—has not been studied extensively. This study explored how Japanese medical students are influenced by interactions with the public as part of the PIF process. Based on these findings, we now discuss the theoretical implications of supporting professional identity development from a different perspective, particularly for students in the pre-stage of legitimate peripheral participation or those transitioning into legitimate peripheral participation. The results emphasize the significant role of public dialogue in fostering PIF and highlight key considerations and interpretations for student identity development. In this study, interviews were conducted with three medical students, which represents a very small sample size. Therefore, rather than aiming for generalizability, this study seeks to provide in-depth insights into the participants' experiences. This study adopts a qualitative research approach grounded in the interpretivist paradigm, utilizing the SCAT method. Unlike grounded theory, which is rooted in the post-positivist, fact-oriented paradigm, SCAT is better suited to interpretivist, meaning-oriented qualitative research and supports studies with a single participant [15,16]. Consequently, the goal was not to achieve theoretical saturation, as in grounded theory, but rather to pursue specific and individual insights through a small number of research participants. This approach enables the extraction of generalizable and universal insights, despite the limited number of participants, reflecting the interdependent nature of human psychological, social, and cultural realities [7]. The aim of SCAT is not the reproducibility of results but rather the contribution of the derived theories to future practice and research in the respective field. To avoid misinterpretation and incorrect theorization when conducting deep analyses with a small number of participants, it is essential to utilize existing theoretical frameworks and construct innovative theories as extensions of these frameworks. This study focuses on PIF, with the theoretical framework proposed by Cruess et al. providing the foundational basis for the comprehensibility and relevance of this research.

Cruess et al. suggested that PIF in medical education can be classified using Kegan's stages of identity development [3]. They identified medical students as primarily situated within stages 2–4 (Imperial, Interpersonal, and Institutional). In line with this framework, a study by Tagawa et al. on Japanese medical students found that elements of Kegan's second stage were the most prominent on average [17], indicating that these students were in the process of socialization.

In this study, the students also described having limited opportunities to recognize or express their professional self, which aligns with previous findings suggesting that medical students are in Kegan's stages 2–3 of socialization.

Cruess et al. further argued that the integration of personal identity and professional identity occurs through participation in CoPs. This study observed a similar phenomenon but only among fifth-year students with clinical training experience. This finding supports the significance of participation in CoPs, highlighting its role in fostering the integration of individual and professional identities.

As mentioned earlier, this study is based on the theoretical framework proposed by Cruess et al. How do these findings align with alternative perspectives on PIF? Varpio et al. analyzed and reported on 43 studies that investigated medical education interventions aimed at influencing PIF [18]. According to their findings, the majority of these studies focused on undergraduate medical education, with the most common approach being reflective writing and the use of narrative reflections. They noted that an individualistic approach predominated in interventions targeting PIF. In their report, they stated that "this study demonstrates that individualist perspectives reliant on reflective writing elide the impact of social contexts of PIF." These findings suggest a prevailing individualistic bias in PIF approaches within undergraduate medical education. Given this tendency, public engagement may serve as a valuable complementary approach, mitigating the limitations of solely individualistic methods in fostering PIF.

Furthermore, this study demonstrated the potential of dialogue with the general public to facilitate reflection on and articulation of participation in such CoPs, thereby supporting the integration of the aforementioned identities. Despite these findings, several aspects require further investigation. In this study, outreach events supported by the university were utilized as opportunities for dialogue with the public. As mentioned in the Results section, discussions about professional identity are difficult to hold in everyday life. Therefore, simply exposing the students to the general public is insufficient. It is essential to create an environment in which students can comfortably talk about the medical education they receive and healthcare in general. In addition, the general public who visited the booths at this event did so with the intention of learning about the current state of medical education directly from the medical students. This, in turn, allowed students to speak candidly about their efforts and learning experience. During the interviews, a fifth-year student remarked, "In conversations with patients during clinical training, I rarely had the opportunity to talk about my identity as a medical student, as I was focused on discussing medical knowledge, patient conditions, and treatments." It is important to note that this benefit stemmed from engaging with the general public, an interaction distinct from their usual exchanges with family and acquaintances or with patients during clinical training.

As previously mentioned in the introduction, medical education differs by country. When applying the findings of this study, attention must be paid to the differences between systems and cultures. As medical education in Japan begins immediately after high school graduation, unlike in the United States, these findings may be particularly applicable to medical students in countries that have similar systems to Japan, such as Germany.

Cross-cultural comparisons of PIF suggest that different societal and educational structures shape how individuals in the medical field perceive and develop their professional identities. For instance, Wahid et al. highlights the significant role of social recognition in shaping the PIF of medical educators in non-Western settings, indicating that societal validation plays a crucial role in professional identity development [19]. Similarly, studies on medical students have shown that in Germany, the emphasis on structured mentorship programs fosters PIF through hierarchical professional relationships, whereas in the U.S., early clinical exposure plays a crucial role in shaping students' professional identities. Our findings suggest that in Japan, public engagement can serve as a unique factor in PIF, offering students an alternative means of professional identity reinforcement outside of the clinical setting.

In addition, this study could not sufficiently examine the peer interactions that occurred when senior and junior students collaborated to manage an event. Senior students expressed that they felt they could "speak naturally based on their own experiences" engaging in dialogue with the general public. Junior students, on the other hand, reflected on their dual role as both learners and educators, describing their experience of "explaining to the general public what they had learned from the senior students." This aspect presents an intriguing topic for further investigation, which will be valuable in future research.

## Conclusion

This study explored how Japanese medical students are influenced by the process of PIF through dialogue with the general public. The findings revealed that medical students who participated in the event where they explained their

experiences as direct recipients of education to the general public had a positive impact on their PIF development. In other words, interaction with the general public, distinct from the professional group they currently belonged to or would be part in the future, was found to counterintuitively heighten students' awareness of their own professional identity. This study suggests that dialogue with the general public, traditionally been undervalued in comparison to interactions within professional groups, can contribute to the socialization process by fostering the reevaluation of professional identity and its integration with personal identity.

## Supporting information

**S1 File. Summary of Japanese References.**
(DOCX)

**S2 File. Storyline and Theoretical Description (English Translation).**
(DOCX)

## Acknowledgments

We thank all the students who participated in the interviews and all the faculty members who facilitated the Academic Day.

## Author contributions

**Conceptualization:** Masashi Ikuno.

**Data curation:** Masashi Ikuno, Akiko Tokinobu, Tomoko Miyoshi.

**Formal analysis:** Masashi Ikuno, Akiko Tokinobu, Tomoko Miyoshi, Hitomi Kataoka.

**Funding acquisition:** Hitomi Kataoka.

**Investigation:** Masashi Ikuno, Akiko Tokinobu, Tomoko Miyoshi.

**Methodology:** Masashi Ikuno, Hitomi Kataoka.

**Project administration:** Masashi Ikuno, Hitomi Kataoka.

**Resources:** Masashi Ikuno.

**Supervision:** Hitomi Kataoka.

**Validation:** Masashi Ikuno, Akiko Tokinobu, Tomoko Miyoshi.

**Visualization:** Masashi Ikuno.

**Writing – original draft:** Masashi Ikuno.

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
