## [Decision Letter · Decision Letter 0]

PONE-D-25-01295

'I'm Working Hard to Become a Doctor’ That Somehow Feels Right to Me Now

– Dialogue with the General Public as an Effective Opportunity for Professional Identity Formation –

PLOS ONE

Dear Dr. Ikuno   , 

Thank you for submitting your manuscript to PLOS ONE. After careful consideration, we feel that it has merit but does not fully meet PLOS ONE’s publication criteria as it currently stands. Therefore, we invite you to submit a revised version of the manuscript that addresses the points raised during the review process.

We look forward to receiving your revised manuscript.

Kind regards,

Anika Sehgal

Academic Editor

PLOS ONE

Journal Requirements:

Reviewers' comments:

Reviewer's Responses to Questions

**Comments to the Author**

1. Is the manuscript technically sound, and do the data support the conclusions?

Reviewer #1: Partly

Reviewer #2: Yes

2. Has the statistical analysis been performed appropriately and rigorously? 

Reviewer #1: N/A

Reviewer #2: N/A

3. Have the authors made all data underlying the findings in their manuscript fully available?

Reviewer #1: No

Reviewer #2: No

4. Is the manuscript presented in an intelligible fashion and written in standard English?

Reviewer #1: No

Reviewer #2: Yes

5. Review Comments to the Author

Reviewer #1: CORRECTIONS

1. Title:

a) The title is quite long and somewhat ambiguous. Consider rewording it for clarity and conciseness. For example: "Dialogue with the Public: A Catalyst for Professional Identity Formation in Medical Students"

b) The phrase "That Somehow Feels Right to Me Now" is informal and might not fit well in an academic setting.

2. Abstract:

a) The abstract states that interviews were conducted with three medical students. This is a very small sample size and might not be sufficient for generalizable conclusions. Consider acknowledging this as a limitation.

b) The abstract describes findings about "Lower-year" and "Upper-year" students but does not provide specific supporting evidence.

c) The phrase "counterintuitively promote" should be replaced with something clearer, e.g., "Unexpectedly enhanced".

3. Introduction:

a) The introduction does not sufficiently justify why dialogue with the general public is a crucial factor in professional identity formation (PIF).

b) The study could benefit from further elaboration on how this research fills a gap in existing PIF literature.

4. Method:

a) Institutional Review Boards (IRBs) usually require signed or documented informed consent. Specify if there was a consent form with a clear explanation of rights.

5. Discussion:

a) The discussion primarily relies on Cruess et al.’s work but does not sufficiently engage with other theoretical models.

b) The comparison between Germany, Japan, and the U.S. is interesting but could be expanded.

6. References:

a) Some citations are incomplete: "Otani T. SCAT: steps for coding and theorization, qualitative data analysis method. English version Last updated 4. Nov. 2015 ed." → Provide the correct DOI or official link.

b) Reference 14 and 15 are in Japanese and may not be accessible to all readers. Consider summarizing key insights in English.

Reviewer #2: Dear authors, it is indeed an interesting study conducted in a unique manner. However, few minor revisions will improve the quality of your manuscript.

For reporting excerpts of the study participants, give identifiers to the participants' ID to be brief and focused. For example fifth year male student can be coded as 5MS where 5 stands for year of study, M stands for sex/gender and S stands for student. I don't like the labelling of students as lower year and upper year students. You can call them senior or junior students or preclinical or clinical students!

Correct me if my understanding is wrong for the following facts:

The public interaction lasted for 5 hours on a single day. The interview was taken once individually face to face with students lasting for 15-20 min each. But the study conduction days are described as three weeks? Please write these facts clearly.

The English used in interview guide needs editing.

What motivated you to participate in the event? What factor/s made you decide to participate?

What were your feelings after participating upfront on Academic day? What were the positive versus challenging aspects of participation.

regarding communication, I didnot understand the question

In personal change, what I have understood is you wanted to ask "Did this experience bring any changes or enhancements in your approach/feeling towards being a doctor?"

In past experiences: Did you have any previous opportunity/ies to engage in a dialogue with the general public like this?

Can you share any experience/s of discussing the medical curriculum or medical practices with friends, family, or others before?

More excerpts can be given to strengthen the authors' claim of PIF by public interaction.

Few studies also mention the role of society in shaping PIF of doctors and medical teachers

I am sharing links.

https://doi.org/10.3389/fmed.2023.1323075

https://doi.org/10.3389/feduc.2024.1307560

10.1080/0142159X.2021.1922657

You must read Amee guide 149 to further strenghthen your study https://doi.org/10.1080/0142159X.2022.2057287

I look forward to read revisd version of your manuscript.

6. PLOS authors have the option to publish the peer review history of their article (what does this mean?). If published, this will include your full peer review and any attached files.

Reviewer #1: **Yes: **SALMAN ASHFAQ AHMAD

Reviewer #2: **Yes: **Faiza Kiran

---

## [Author Response · Author response to Decision Letter 1]

7 Mar 2025

Response to Reviewer 1:

We wish to express our appreciation to the reviewer for these insightful comments regarding our paper. Below, we address each of your points in detail:

Comments:

1. Title:

a) The title is quite long and somewhat ambiguous. Consider rewording it for clarity and conciseness. For example: "Dialogue with the Public: A Catalyst for Professional Identity Formation in Medical Students"

b) The phrase "That Somehow Feels Right to Me Now" is informal and might not fit well in an academic setting.

Response:

Thank you for your suggestion. We have revised the title for clarity and conciseness. The new title is:

"Dialogue with the Public: A Catalyst for Professional Identity Formation in Medical Students."

We removed the phrase "That Somehow Feels Right to Me Now" to ensure a more academic tone.

Comments:

2. Abstract:

a) The abstract states that interviews were conducted with three medical students. This is a very small sample size and might not be sufficient for generalizable conclusions. Consider acknowledging this as a limitation.

Response:

Thank you for your insightful comment. We acknowledge that the sample size of three students is small and may limit the generalizability of our findings. In response to your suggestion, we have explicitly stated this limitation in the Discussion section.

On page 17 in the Discussion Section:

In this study, interviews were conducted with three medical students, which represents a very small sample size. Therefore, rather than aiming for generalizability, this study seeks to provide in-depth insights into the participants’ experiences.

Comments:

2. Abstract:

b) The abstract describes findings about "Lower-year" and "Upper-year" students but does not provide specific supporting evidence.

c) The phrase "counterintuitively promote" should be replaced with something clearer, e.g., "Unexpectedly enhanced".

Response:

We appreciate your feedback on this point. To address your concern, we have revised the abstract to include specific supporting evidence derived from our SCAT analysis. We now explicitly describe how preclinical students (third-year students) reaffirmed their professional identity through empathetic understanding of event attendees, while clinical students (fifth-year students) disclosed their professional identity during dialogues with the public, leading to its integration with their personal identity. Additionally, we have incorporated direct quotes from the students, which illustrate these findings more concretely. We also have revised this phrase to "unexpectedly enhanced" to improve clarity and readability.

Results

The students reflected on their dialogue with the general public about medical education and discussed infrequent opportunities to recognize or disclose their professional identity in daily life. A preclinical student (third-year) noted, "I felt like I shared a similar sense of amazement with the visitors" and "It’s obvious, but I really came to realize that I’m seen from the perspective of a medical student or a healthcare professional." In contrast, a clinical student (fifth-year) reflected, "Sharing my feelings about whether I’ve been helping others during my clinical training felt closely tied to me—there wasn’t much of a gap." SCAT analysis revealed that preclinical students emphasized 'reaffirming their professional identity through empathetic understanding,' while clinical students focused on 'disclosing their professional identity, leading to its integration with their personal identity.'

Conclusion:

Dialogue with the general public unexpectedly enhanced medical students' awareness of their professional identity. In the process of professional identity formation, dialogue with the general public can support the traditionally emphasized socialization process through participation in professional groups, particularly by facilitating the re-recognition of professional identity and its integration with personal identity.

Comments:

3. Introduction:

a) The introduction does not sufficiently justify why dialogue with the general public is a crucial factor in professional identity formation (PIF).

b) The study could benefit from further elaboration on how this research fills a gap in existing PIF literature.

Response:

Thank you for your valuable feedback on the Introduction section. We have revised this section to more clearly justify the significance of dialogue with the general public in professional identity formation (PIF). We believe these revisions directly address the concerns raised and enhance the clarity of our research justification. We appreciate your insightful comments, which have helped improve our manuscript.

On page 5 in the Introduction Section:

Thus, PIF is traditionally understood as a process of socialization within professional communities, particularly through interactions with mentors, peers, and clinical experiences. While these internal factors have been extensively studied, PIF is also influenced by broader social interactions, including engagement with the general public. Recent studies suggest that broader social interactions, including engagement with the general public, also contribute significantly to PIF [5,6]. These reports showed that public engagement can provide opportunities to articulate professional roles, receive external validation, and reflect on their evolving identities. However, as mentioned earlier, most existing literature on PIF has focused on socialization within medical communities, leaving a gap in understanding how external interactions shape identity development. This study aims to address this gap by investigating how dialogue with the general public influences the PIF process among Japanese medical students.

To explore this, we conducted a qualitative study using semi-structured interviews with medical students who participated in an interactive public event at Kyoto University. During the event, titled "The Past and Future of Medical Education," students introduced the public to the types of classes and training involved in medical education. By analyzing students' reflections on this outreach activity, we examined how public engagement complements traditional forms of professional socialization. A thematic analysis was conducted using Steps for Coding and Theorization (SCAT).

Comments:

4. Method:

a) Institutional Review Boards (IRBs) usually require signed or documented informed consent. Specify if there was a consent form with a clear explanation of rights.

Response:

Thank you for your comment. In response, we have clarified our consent process in the Methods section. Participants were provided with a detailed written explanation of the study and were asked to confirm their consent through a web form approved by the IRB. This process included reviewing the consent document, checking an agreement checkbox, and entering their name to confirm consent.

On page 7 in the Method Section:

Consent was obtained in accordance with the IRB-approved format and procedure through a web form. Participants were required to review the consent document, check an agreement checkbox, and enter their name to confirm their consent.

Comments:

5. Discussion:

a) The discussion primarily relies on Cruess et al.’s work but does not sufficiently engage with other theoretical models.

Response:

Thank you for your insightful comment regarding the theoretical grounding of our discussion. In response, we have incorporated additional engagement with alternative perspectives on PIF. We believe this addition strengthened our supposition by situating our findings within a broader theoretical landscape and considering alternative perspectives on PIF.

On page 19 in the Discussion Section:

As mentioned earlier, this study is based on the theoretical framework proposed by Cruess et al. How do these findings align with alternative perspectives on PIF? Varpio et al. analyzed and reported on 43 studies that investigated medical education interventions aimed at influencing PIF [18] . According to their findings, the majority of these studies focused on undergraduate medical education, with the most common approach being reflective writing and the use of narrative reflections. They noted that an individualistic approach predominated in interventions targeting PIF. In their report, they stated that "this study demonstrates that individualist perspectives reliant on reflective writing elide the impact of social contexts of PIF." These findings suggest a prevailing individualistic bias in PIF approaches within undergraduate medical education. Given this tendency, public engagement may serve as a valuable complementary approach, mitigating the limitations of solely individualistic methods in fostering PIF.

Comments:

b) The comparison between Germany, Japan, and the U.S. is interesting but could be expanded.

Response:

Thank you for your valuable feedback regarding the cross-cultural comparison between Germany, Japan, and the U.S. In response, we have expanded this section by incorporating additional perspectives on how different societal and educational structures shape professional identity formation (PIF). Specifically, we have included Wahid’s (2021) discussion on the role of social recognition in PIF, particularly in non-Western settings, highlighting the importance of societal validation. We believe these additions provide a more nuanced comparative analysis and appreciate your suggestions, which have helped us strengthen this discussion.

On page 21 in the Discussion Section:

Cross-cultural comparisons of PIF suggest that different societal and educational structures shape how individuals in the medical field perceive and develop their professional identities. For instance, Wahid et al. highlights the significant role of social recognition in shaping the PIF of medical educators in non-Western settings, indicating that societal validation plays a crucial role in professional identity development [19]. Similarly, studies on medical students have shown that in Germany, the emphasis on structured mentorship programs fosters PIF through hierarchical professional relationships, whereas in the U.S., early clinical exposure plays a crucial role in shaping students’ professional identities. Our findings suggest that in Japan, public engagement can serve as a unique factor in PIF, offering students an alternative means of professional identity reinforcement outside of the clinical setting.

Comments:

6. References:

a) Some citations are incomplete: "Otani T. SCAT: steps for coding and theorization, qualitative data analysis method. English version Last updated 4. Nov. 2015 ed." → Provide the correct DOI or official link.

b) Reference 14 and 15 are in Japanese and may not be accessible to all readers. Consider summarizing key insights in English.

Response:

Thank you for your valuable comment. Upon further review, we have determined that the reference to Otani’s SCAT English version website was not an appropriate academic citation, as it primarily serves as an explanatory resource rather than a formal publication. In response, we have removed this citation from our references. In addition, we have added a supplemental material file summarizing the key insights of References 5, 14, and 15, as these references are available in Japanese only. Thank you for your constructive feedback, which has helped us improve the clarity and accessibility of our manuscript.

Response to Reviewer 2:

Thank you for your thoughtful and constructive feedback. We appreciate your positive comments on our study's uniqueness and have carefully considered your suggestions for improving the manuscript. Below, we address each of your points in detail:

Comments:

For reporting excerpts of the study participants, give identifiers to the participants' ID to be brief and focused. For example fifth year male student can be coded as 5MS where 5 stands for year of study, M stands for sex/gender and S stands for student. I don't like the labelling of students as lower year and upper year students. You can call them senior or junior students or preclinical or clinical students!

Response:

Thank you for your valuable suggestion regarding participant identifiers. In response, we have incorporated the identifier system as advised. We now use the format **[Year of study] [Sex/Gender] [Student], such as 5MS for a fifth-year male student. Additionally, we have revised the terminology throughout the manuscript, replacing "Lower-year" and "Upper-year" with "Preclinical" and "Clinical" students to ensure clarity and consistency.

Comments:

For reporting excerpts of the study participants, give identifiers to the participants' ID to be brief and focused. For example fifth year male student can be coded as 5MS where 5 stands for year of study, M stands for sex/gender and S stands for student. I don't like the labelling of students as lower year and upper year students. You can call them senior or junior students or preclinical or clinical students!

Correct me if my understanding is wrong for the following facts:

The public interaction lasted for 5 hours on a single day. The interview was taken once individually face to face with students lasting for 15-20 min each. But the study conduction days are described as three weeks? Please write these facts clearly.

Response:

Thank you for your careful review of our study timeline. The public interaction event lasted 5 hours on a single day as part of Academic Day 2024. Individual interviews with participants were conducted once, face-to-face, and generally lasted around 20 minutes. The three-week study period refers to the total duration from the public interaction event to the completion of all interviews. To ensure clarity, we have revised the Methods section to explicitly state this timeline. Thank you for highlighting this point.

On page 6 in the Method Section:

The recruitment period for this study was from September 17 to October 31, 2024. The total study period lasted three weeks, covering the time from the public interaction event to the completion of all interviews and data processing. The public interaction event itself took place on a single day, lasting five hours. After the event, individual interviews were conducted once per participant.

Comments:

The English used in interview guide needs editing.

What motivated you to participate in the event? What factor/s made you decide to participate?

What were your feelings after participating upfront on Academic day? What were the positive versus challenging aspects of participation.

regarding communication, I didnot understand the question

In personal change, what I have understood is you wanted to ask "Did this experience bring any changes or enhancements in your approach/feeling towards being a doctor?"

In past experiences: Did you have any previous opportunity/ies to engage in a dialogue with the general public like this?

Can you share any experience/s of discussing the medical curriculum or medical practices with friends, family, or others before?

Response:

Thank you for your valuable feedback on the clarity of the interview guide. We appreciate your suggestions and have revised the questions accordingly to ensure clearer and more precise wording. We sincerely appreciate your detailed suggestions, which have helped improve the quality and clarity of our interview guide.

1. Feelings When Deciding to Participate in Academic Day 2024

·What motivated you to participate in the event?

·What factor/s made you decide to participate?

2. Impressions of Participation

·What were your feelings after participating upfront on Academic day?

·What were the positive versus challenging aspects of participation.?

3. Communication

·Through this experience, did you gain any new insights or awareness about communication?

4. Personal Changes

·Did this experience bring any changes or enhancements in your motivation for studying and training in medical school?

5. Teamwork

·What are your thoughts on working together with other students during this event?

6. Past Experiences in Dialogue with the General Public

-Did you have any previous opportunity/ies to engage in a dialogu

---

## [Decision Letter · Decision Letter 1]

Dialogue with the Public: A Catalyst for Professional Identity Formation in Medical Students.

PONE-D-25-01295R1

Dear Dr. Ikuno,

We’re pleased to inform you that your manuscript has been judged scientifically suitable for publication and will be formally accepted for publication once it meets all outstanding technical requirements.

Kind regards,

Emily Lund

Academic Editor

PLOS ONE

Additional Editor Comments (optional):

I believe that the authors have adequately addressed all reviewer comments, and the manuscript is now ready for publication.

Reviewers' comments:

Reviewer's Responses to Questions

**Comments to the Author**

1. If the authors have adequately addressed your comments raised in a previous round of review and you feel that this manuscript is now acceptable for publication, you may indicate that here to bypass the “Comments to the Author” section, enter your conflict of interest statement in the “Confidential to Editor” section, and submit your "Accept" recommendation.

Reviewer #1: All comments have been addressed

Reviewer #2: All comments have been addressed

2. Is the manuscript technically sound, and do the data support the conclusions?

Reviewer #1: Yes

Reviewer #2: Yes

3. Has the statistical analysis been performed appropriately and rigorously? 

Reviewer #1: Yes

Reviewer #2: N/A

4. Have the authors made all data underlying the findings in their manuscript fully available?

Reviewer #1: Yes

Reviewer #2: Yes

5. Is the manuscript presented in an intelligible fashion and written in standard English?

Reviewer #1: Yes

Reviewer #2: Yes

6. Review Comments to the Author

Reviewer #1: (No Response)

Reviewer #2: In abstract section, last line instead of writing "re-recognizing identity" use "reaffirming identity".

In last paragraph of introduction section, write research questions and objectives, instead of talking about methodology.

7. PLOS authors have the option to publish the peer review history of their article (what does this mean?). If published, this will include your full peer review and any attached files.

Reviewer #1: **Yes: **Salman Ashfaq Ahmad

Reviewer #2: **Yes: **Faiza Kiran

---

## [Editor Report · Acceptance letter]

PONE-D-25-01295R1

PLOS ONE

Dear Dr. Ikuno,

I'm pleased to inform you that your manuscript has been deemed suitable for publication in PLOS ONE. Congratulations! Your manuscript is now being handed over to our production team.

Kind regards,

on behalf of

Dr. Emily Lund

Academic Editor

PLOS ONE